# From Cell Clones to Recombinant Protein Product Heterogeneity in Chinese Hamster Ovary Cell Systems

**DOI:** 10.3390/ijms26031324

**Published:** 2025-02-04

**Authors:** Chong Wang, Xiao Guo, Wen Wang, Jia-Xin Li, Tian-Yun Wang

**Affiliations:** 1School of Medical Engineering, Xinxiang Medical University, Xinxiang 453003, China; wagnchogn@163.com; 2School of Life Science and Technology, Xinxiang Medical University, Xinxiang 453003, China; guoxiaowch@126.com (X.G.); 15027466716@163.com (J.-X.L.); 3International Joint Laboratory of Recombinant Pharmaceutical Protein Expression System of Henan, Xinxiang Medical University, Xinxiang 453003, China; 241025@xxmu.edu.cn

**Keywords:** Chinese hamster ovary cells, protein of interest, clone aggregation, protein aggregation, culture process

## Abstract

Chinese hamster ovary (CHO) cells are commonly used to produce recombinant therapeutic proteins (RTPs). The yield of RTPs in CHO cells has been greatly improved through cell editing and optimization of culture media, cell culture processes, and expression vectors. However, the heterogeneity of cell clones and product aggregation considerably affect the yield and quality of RTPs. Recently, novel technologies such as semi-targeted and site-specific transgene integration, endoplasmic reticulum-residents, and cell culture process optimization have been used to address these issues. In this review, novel developments in the field of CHO cell expression system heterogeneity are summarized. Moreover, the advantages and limitations of the new strategies are discussed, and important methods for the control of RTP quality are outlined.

## 1. Introduction

Recombinant therapeutic proteins (RTPs), including therapeutic antibodies, are currently among the fastest growing and most profitable products in the pharmaceutical market. Based on recent data, the global biopharmaceutical market size in 2021 was 336.6 billion US dollars, of which monoclonal antibody drugs accounted for 205.4 billion US dollars (up to 61.02%), and it is expected to increase to 292.1 billion US dollars by 2025. *Escherichia coli*, yeast, insect cells, and mammalian cells can be used to manufacture RTPs. Because the post-translational modifications (PTMs) of mammalian cells are similar to those of human cells, 81% of recombinant therapeutic antibodies newly approved during 2018–2022 were derived from them, making them an important platform for producing RTPs [1]. The preferred mammalian cells include the Chinese hamster ovary (CHO), human embryonic kidney 293, baby hamster kidney, Sp2/0 myeloma, and NS0 mouse myeloma cells. CHO cells are among the most favored systems in the industrial production of RTPs [1].

CHO cells can be cultured in suspension, at high density, and on a large scale (up to 12,000 L or more). They offer easy integration and expression of transgenes, production of proteins with PTMs similar to those in human cells, and resistance to human viral infections [2,3]. Owing to these advantages, CHO cells have become the primary RTP expression system. Recently, through strategies such as optimization of culture media, cell culture processes, and expression vectors, the yield of RTPs in CHO cells has greatly improved. Antibody production has reached 5.0 g/L and can reach over 13.0 g/L when batch-cultured [4]. Irrespective of the current CHO cell expression system, issues such as the heterogeneity of the cell clone phenotype and product aggregation occur, which seriously influence the yield and quality of RTPs and should urgently be addressed.

## 2. Formation and Strategies to Address Cell Clone Heterogeneity

### 2.1. CHO Cell Line Development

The recombinant CHO cell line development (CLD) process includes transgene cloning and vector construction, transfection and cell pool screening, stable cell clone screening, cell clone expansion, cell bank construction, and culture process optimization (Figure 1). Current methods for cell clone screening mainly include limited dilution (cell density of <0.5 cell/cell), semi-solid medium combined with fluorescence-activated cell sorting screening, and vector combined with semi-solid and fluorescence screening [5]. Currently, recombinant proteins are produced through random transgene integration (RTI) onto the host cell chromosome. Generally, restriction endonucleases are used to linearize expression plasmids and integrate genes of interest (GOIs) and selective genes into the host cell genome [6]. Currently, RTI remains the classic approach for screening stable CHO-producing cell lines due to the lack of more feasible methods which can be simplified and effectively used to generate highly expressed cell clones. However, this approach primarily relies on chance, requiring complex and time-consuming cloning selection and identification of workflows to generate stable and high-performance cell clones [7,8]. A transposase-mediated transgene semi-targeted integration (STI) system has been used in CHO cell line screening [6]. Sleeping Beauty, PiggyBac, Leap in ®, and DirectedLuck ™ Transposition enzyme systems are commonly used transposase systems in biopharmaceuticals [6]. Three transgene integration patterns are listed in Table 1.

### 2.2. Transgene Integration Pattern Cause Cell Clone Heterogeneity

Owing to the RTI of GOIs, RTI-mediated cell cloning results in a highly heterogeneous cell pool with considerable interclonal variation and highly diverse phenotypes [9]. Transgenes mediated by RTI that are integrated at several sites can result in downregulated, upregulated, or completely silenced gene sets. This phenomenon can lead to clonal heterogeneity of cells and can simultaneously produce high-yield, fast-growing clonal cell lines [7]. Regardless of the type of RTI or STI system, differences in transgene expression levels and stability exist between different cell clones, resulting in phenotypic heterogeneity of cell clones [10] as well as phenotypic heterogeneity between different subclones derived from one identical cell line [11,21]. Only 1%–2% of cell clones produced by RTI can stably and efficiently express recombinant proteins suitable for large-scale industrial production [12]. Thousands of single cells must be selected to screen clonal cell lines with increased cellular productivity and growth and an adequate product quality profile [8]. Lee et al. showed that 210 cell clones were selected and cultured at 37 °C, with yields of the protein of interest (POI) ranging 1.2–431.3 mg/L [22]. The cultivation temperature lowered to 33 °C enhanced the antibody expression level, but the heterogeneity of the clones also increased, with 210 clones ranging 2.1–2272.0 mg/L. This study also showed that for a highly expressed cell clone (125.0 mg/L) obtained through screening after 35 days of passage, the antibody expression levels differentiated accordingly, ranging 20.3–180.0 mg/L. The cell clones with the highest POI expression were selected for passage for 55 days, the heterogeneity was further increased with antibody expression levels ranging 15.3–280.6 mg/L, and the expression levels of adalimumab exhibited heterogeneity. Among the 72 selected cell clones, the highest expression level was 793.1 mg/L, whereas some single cells showed no expression [8].

By comparing the CHO cell clones produced by the PiggyBac transposase system and RTI, we determined that the transposase system can achieve high copy numbers without amplification and enhanced integration of single but complete GOI cassettes. The clones generated through RTI often contain fragmented and concatemeric integrations, and the phenotypic heterogeneity of cell clones produced by STI is more significant than that of RTI in CHO cells; however, better productivity stability has been observed [6,13].

The factors and molecular mechanisms affecting clonal heterogeneity are believed to be caused by the RTI or STI of the transgenes. The RTI or STI of GOIs in cell chromosomes and their integration into different genomic structural sites can affect POI yields and long-term stability, resulting in phenotypic heterogeneity of cell clones. Transgene integration into regions close to negative regulatory elements or heterochromatin is not conducive to gene expression, resulting in silencing or low gene expression levels, whereas the integration of transgenes into the euchromatin region facilitates efficient expression of transgenes, with higher levels of transgenic expression [14]. CHO cells cultured over a long period show a high incidence of chromosomal abnormalities and transcription errors caused by DNA template mutations [15].

Site-specific integration (SSI) can be used to clone transgenes into predefined gene loci to reduce cell clone heterogeneity in CHO cells. At present, four primary gene editing methods are used for SSI systems: meganuclease, zinc finger nuclease (ZFN), transcription activator-like effector nuclease, and CRISPR/Cas9 systems [10,16,17,18,19,20,23]. CRISPR/Cas9 is widely used to construct stable recombinant CHO cell lines because of its unique targeting mechanism and high cutting efficiency. Several systems have been effectively applied to develop recombinant cell lines through recombinase-mediated cassette exchange in CHO cells, including the Fpl FRT [19], Bxb1 recombinase [24], PhiC31 integrase [19], and Cre-loxP systems [25].

The construction of stable cell clones through SSI requires the identification of highly active and stable genomic sites, which is laborious, difficult, and cost-intensive [6,26]. Several hotspots have been identified, with the most prominent being the hypoxanthine phosphoribosyltransferase (*Hprt1*) gene [27,28]. Compared to the RTI and STI methods, the CLD method based on SSI can integrate transgenes into predefined genomic hotspots, shorten the CLD screening time, and thereby reduce cell clone heterogeneity [6]. However, the low copy number, low expression level, and off-target nature of SSI limit its application in industrial production.

At present, SSI is no longer limited to the integration of individual GOIs, but integrates landing pads to produce stably transfected cell lines that contain selection markers and recognition sites for recombinases or integrases, thereby enabling precise cloning of any category and number of GOI expression cassettes into developed locations [12].

### 2.3. Cell Clone Heterogeneity Is Related to the CHO Cell Line

Normal CHO cells have 22 chromosomes, whereas CHO-K1 cells have only 20 chromosomes [29], of which only 8 are identical. The remaining 12 contain varying degrees of chromosomal variations, such as deletions, translocations, and rearrangements [30]. The CHO-DG44 and CHO-S cells also exhibit varying degrees of chromosomal variation [30]. Because of genomic instability, the initially selected high-expression cell clones exhibit genotype and phenotype drift with prolonged culture time, leading to various product titers and qualities [31]. Treatment of the dihydrofolate reductase gene-deficient cell line CHO DG44 with methotrexate can cause chromosomal rearrangements in CHO cells, resulting in considerable genetic heterogeneity. In addition to the inherent instability of chromosomes, CHO cells exhibit heterogeneity in chromosome number, which results in changes in gene copy number [32].

Genomic analyses of CHO cells revealed that cells without chemical mutagenesis had stable chromosomes and that CHO-K1 cells were a promising option. However, CHO-K1 cells had a high proportion of haploid chromosomes, and the CHO-GS line obtained through zinc finger nuclease (ZFN) gene editing was more commonly used in industrial production [31].

CHO cell lines can be effectively modified to enhance their anti-apoptotic ability. Early apoptotic genes are present in CHO cell cultures [33]; therefore, downregulating the expression of apoptotic genes and overexpressing anti-apoptotic genes can enhance the anti-apoptotic ability of CHO cells. Knocking out Bcl-2-associated protein expression and Bcl-2 antagonists can effectively enhance the anti-apoptotic ability of CHO cells to extend culture time [34]. Overexpression of B-cell lymphoma XL can effectively inhibit cell apoptosis [35], whereas other inhibitors, such as B-cell lymphoma-2, X-linked inhibitor of apoptosis protein, caspase activation inhibitor, and Fas apoptosis inhibitory molecule, can also enhance cell anti-apoptotic ability [33,36]. Moreover, by increasing the growth rate of the cells, enhancing the stress resistance of CHO can prolong the cultivation time. Overexpression of valosin-containing protein can regulate the high cellular growth rate of CHO cells [37], and the co-expression of cyclin-dependent kinase 3 and cytochrome C oxidase subunit 15 genes can increase viable cell density [38].

### 2.4. Transcriptional Silencing Contributes to Transgene Expression Instability

Transcriptional silencing of transgenes causes instability in recombinant proteins. The transcriptional level of cells is affected by DNA methylation, nucleosome localization, histone modification and variation, transcriptional complex binding, and non-coding RNA. The copy number of CHO cells did not change during long-term culture; however, the productivity decreased [8]. Currently, commonly used promoters such as SV40, CMV, and EF-1α typically contain a large CG region that is easily methylated and silenced [39]. CpG island free promoters can reduce early transcriptional silencing but do not enhance long-term expression stability [40]. A CMV promoter containing core CpG island elements can improve the stability of POIs in CHO cells [41]. Histone modifications also cause GOI gene silencing [42,43,44]. Moreover, histone acetylation in cells with unstable expression gradually decreases, indicating that the stability of recombinant proteins is influenced by epigenetics [45]. Instability caused by cell death can occur during continuous high-density cell culture. Temperature, pH, osmotic pressure, dissolved oxygen, nutrients, and toxic metabolites can induce cell death [46,47,48]. Annexin V and Caspase 3 protein overexpression can also induce apoptosis [49]. The host protein hydrolysate released by the lysis of dead cells binds to the extracellular secretion of POI, leading to product inactivation or the formation of harmful substances that affect product quality [50]. Excessive proteases and glycosidases also influence the POI quality [51].

Several chromatin-modifying elements, such as the matrix attachment region (MAR), ubiquitous chromatin opening element (UCOE), insulator, and stabilizing anti-repressor elements, incorporated into an expression vector can buffer DNA integration-dependent inhibition or negative “chromatin location effects” to increase transgene expression and stability [52,53]. MARs can reduce clonal variability when combined to the promoter and enhancer region [54,55]. S4, S10, and 1–68 MAR can reduce the variability in transgene production in CHO cells [53]. Human beta-globin MARs can reduce the variations in CAT expression among various stably transfected cell lines [56]. In the PiggyBac transposon system, MAR 1–68 and X-29 can enhance the GOI expression level per integration; however, the total number of transposition events are not affected [6,57]. Notably, 59-HS4 chicken β-globin can promote GOI expression in CHO cells [58,59]. Locus control regions and UCOEs enhance GOI transcription and stability [60]. Moreover, synthetic promoters and CMV promoter mutations can reduce the tendency for GOI transcriptional silencing [61,62,63].

## 3. Bottleneck of Recombinant Protein Aggregation and Solving Strategies

### 3.1. Endoplasmic Reticulum (ER)–Golgi Inefficiencies Cause Protein Aggregation

Protein aggregation often occurs during the production of RTPs, which consist of six types with different molecular weights [64,65]. Protein aggregation is unsatisfactory because it can considerably affect the safety and efficacy of RTPs and also form secretory obstacles [66,67]. In CHO cells, the level of bispecific antibody aggregates was higher than that of monoclonal antibodies [68]. An aggregate content of 10% can trigger an immune response in humans and even lead to death [69]. Previous studies showed that aggregated interleukin-2 insulin, factor VIII, and interferon (IFN)-α2 can induce an adverse immune response [66,70]. Although aggregated proteins can typically be removed through downstream processing, this can lead to a decrease in protein recovery, affect stability, and limit product shelf life [65].

The precise mechanism underlying protein aggregation remains unclear. The biosynthesis, processing, and subsequent secretion of POIs involves the transcription of genetic information from DNA into precursor mRNA in the nucleus, processing, adding caps and polyA tails, and cleaving introns to form mature mRNA. Mature mRNA is then transported to the cytoplasm and binds to ribosomes, and proteins begin to translate. Signal recognition particles recognize signal peptides, form complexes with newly synthesized peptides, and transport them to the ER cavity, where the signal peptides are cleaved by signal peptidases [71]. Next, protein folding and PTM are performed in the ER cavity, sorted and conveyed to the Golgi body, and then secreted by vesicles into the extracellular space, ultimately forming active proteins [72]. Recombinant proteins are primarily secreted extracellularly in mammalian cells, and the RTP yields by CHO expression systems have been remarkably increased by technological improvements. However, these high concentrations of heterologous proteins can affect cell folding, making it difficult for proteins to fold correctly and even cause aggregation to form various types of aggregates.

The ER of CHO cells has a limited ability to fold, process, and assemble proteins, resulting in inefficient processing of elevated loads of r–mRNA [73]. These inefficiencies cause the accumulation of misfolded or partially unfolded peptides in the ER cavity, which are eliminated by the ER-associated degradation (ERAD) pathway [74]. When excessively misfolded or partially unfolded peptides exceed the ERAD clearance capacity, they decrease ER function and initiate the unfolded protein response, leading to aggregation [74,75,76,77]. Transport from the ER to the Golgi apparatus is the rate-limiting step in protein synthesis and secretion. Delayed transport or an insufficient number of cargo receptors can affect transport and lead to aggregation [78,79].

The upregulation of PDI, BiP, Calnexin, and Ero1l ER-resident chaperones can reduce protein aggregation [67,80]. Hsp105α and Hsp105β inhibits the protein aggregation through the induction of Hsp70 [81]. However, increased UPR levels may induce apoptosis and affect clone stability [82].

ER–Golgi intermediate compartment 53 and multiple coagulation factor deficiency protein 2 increased antibody production in CHO cell batch cultures. In cells overexpressing this receptor, antibody expression increases and aggregation decreases [78]. The intracellular mechanisms and strategies for overcoming recombinant protein aggregation are listed in Table 2.

### 3.2. Molecular Structures Lead to the Misfolding and Aggregation

The primary, secondary, and tertiary structures of proteins and their glycosylation patterns are susceptible to chemical damage. The characteristics of amino acids and the high entropy and low entropy of newly synthesized peptides and natural protein forms can influence the correct protein folding states [83,84]. Gene mutations, transcription or translation errors, the collapse of folding and chaperone mechanisms, incorrect PTMs, protein transport, and environmental changes can cause structural changes in proteins, thus leading to protein misfolding [85]. Exposed hydrophobic amino acid residues or partially folded proteins can cause their natural state to become unstable, leading to the formation of protein aggregates. High hydrophobicity, a tendency to form B-sheet secondary structures, and low net charge drive high protein aggregation in polypeptide chains [74,86]. Monoclonal antibodies (mAbs) can form different types of aggregates, and the aggregation of bispecific antibodies expressed in CHO cells is higher than that of mAbs [68].

Although the mechanism underlying protein aggregation remains unclear, reasonable changes in protein sequence and structure can effectively reduce aggregation [87]. Interference with or disruption of the structure of unfolded monomers stabilizes natural monomers or reduces the stability of partially folded monomers. Altering the surface charge of partially folded proteins to enhance their repulsion can reduce aggregates and the aggregation of Ang2mAb through mutation of Cys49 to Thr or Asn [88,89].

To address the issue of BsAbs aggregation, several measures have been developed to prevent the heterodimerization of two different heavy chain (HC) species and the differences in their light chain (LC)/HC interactions, including CrossMab technology for LC and HC region exchange, orthogonal Fab interface technology for transforming antibody Fab interfaces, bispecific antibodies through protein transport, tcBsIgG technology for connecting LC variable region to HC through G4S linker, LUZ-Y technology of adding leucine zipper at the C-terminus of antibody CH3 domain, and OAscFab IgG technology [65,90,91]. Technologies involving the HC primarily include knobs-into-holes technology, Fab arm exchange technology, and DEKK mutation of CH3 electrostatic modification [65]. These technologies can improve the quality of BsAbs by advancing the correct pairing of HC/LC and minimizing aggregation to the greatest extent possible. Changing the ratio of LC to HC can promote the correct folding of antibodies [92]. The LC:HC ratio was regulated by vector elements to enhance the quantity of LC or to reduce HC, thereby reducing aggregation [62,93].

### 3.3. Unequal Expression or Insufficient Secretion

An inappropriate ratio of LC to HC during antibody assembly can cause the antibodies to fail to fold correctly and produce aggregates. Excessive HC or limited molecular chaperones lead to mAb product aggregation in CHO cells [94]. Protein secretion depends on signal peptides, and different signal peptide sequences exhibit varying secretion efficiencies for heterologous proteins [95]. Inappropriate signaling peptides can lead to poor protein secretion and misfolding, thus resulting in aggregations [96].

### 3.4. ER Autophagy Dysfunction

Transcriptome analysis of bispecific antibody cell lines with high and low aggregates have revealed 421 differentially expressed genes, of which 127 were significantly upregulated and 67 were significantly downregulated. Among the upregulated genes, *AKT1S1*, *PIK3C3*, *ATG101*, *LAMP2*, *TOLLIP*, and *ATG16LL* are ER autophagy genes [97,98,99]. Overexpression of *AKT1S1* and *ATG16L1* decreased the formation of antibody aggregates, indicating that ER autophagy can eliminate unfolded protein reactions and degrade protein aggregates, facilitating the correct folding of proteins in the ER [16].

### 3.5. Extracellular Environment

Although the intrinsic properties of proteins are considered the primary reason for protein aggregation, culture conditions such as temperature, pH, and shear stress can also enhance protein aggregation levels [100]. The cell culture environment affects protein aggregation. For example, the reducing agent cysteine can be oxidized by metals, producing free radicals that modify proteins and cause aggregation [101]. By optimizing the composition of the medium and process parameters, reducing temperature and pH, increasing osmotic pressure, and dissolving oxygen, the antibody yields can be increased and aggregation can be reduced [77,102] (Table 3).

Optimizing the composition of the culture medium can increase expression levels and reduce the formation of aggregates. Previous studies have shown that trehalose can reduce the HMW of BsAbs by two-thirds and aggregate formation in mAbs [105,113]. Tyrosine, an essential amino acid, maintains antibody expression; however, high concentrations can also lead to high aggregation [103]. Tehalose, proline, glycerol, dimethyl sulfoxide, sodium 4-phenylbutyrate, sodium butyrate, and choline can reduce the aggregation of recombinant antibodies [104,106,107,114].

Optimization of culture process parameters can also significantly decrease antibody aggregation. At temperatures below 37 °C, recombinant protein productivity can be improved in CHO cells [115], and the reduced aggregation of INF and tumor necrosis factor receptor-Fc has also been observed [108]. However, mAb aggregate formation can also be enhanced when lowering the culture temperature to 31.5 °C because of the limited folding capacity in the ER [109]. Protein aggregation can also be affected by pH, and aggregation of the Flag-tagged cartilage oligomeric matrix protein angiopoietin-1 is significantly reduced at pH 7.5 [110]. Hyperosmolality, caused by NaCl addition, can reduce the cartilage oligomeric matrix protein-angiopoietin-1 (OMP-Ang1) and IFN-β aggregation [111,112]; however, when osmolality is elevated by adding sorbitol, the OMP-Ang1 aggregates cannot be decreased [111]. During perfusion culture, mitochondrial dysfunction-induced glutathione oxidation and ER stress can create a favorable intracellular environment, reduce BsAb antibody aggregation, and increase volumetric yields [65,116,117].

## 4. Conclusions and Future Perspectives

Owing to the complexity of RTPs, numerous factors affect their characteristics, and they exhibit a certain degree of heterogeneity during production, including the effects of intracellular protein synthesis and environmental factors, even in different batches. Despite heterogeneity, all RTPs should maintain consistency in terms of quality and clinical performance. Understanding the heterogeneity in RTP production is crucial for producing high-quality therapeutic proteins. RTI is a classic transgenic integration method; however, it involves severe cell clone heterogeneity, unstable protein expression, and the time-consuming and laborious screening of high-quality cell clones. The application of STI technologies in CHO cell cloning has increased expression levels and reduced workload; however, heterogeneity exists in cell cloning. SSI technology can address cell clone heterogeneity; however, the low copy number, low expression level, and off-target nature of SSI limit its application in industrial production. To address these issues, studies have combined new gene editing techniques with transposases, such as Cas9 nucleases fused with transposases, to enable transposases to be directed to specific locations in the host cell genome [118,119].

Aggregation negatively affects the yield, shelf life, efficacy, and safety of RTPs in CHO cells. Inefficiencies in the ER–Golgi, the molecular structures of RTPs, and the extracellular environment can cause intracellular and extracellular aggregation of RTPs. Understanding the processes of intracellular protein secretion, folding, and aggregation, cell editing, overexpression of ER-related molecules, modification of gene structures, and optimization of the culture environment can help reduce the production of aggregates. With the use of multi-omics data and AI technology, understanding of the gene structure and genetic information of CHO cells, and the establishment of large-scale industrial recombinant protein technology, the heterogeneity of CHO cell expression systems can be significantly reduced.

## Figures and Tables

**Figure 1 ijms-26-01324-f001:**
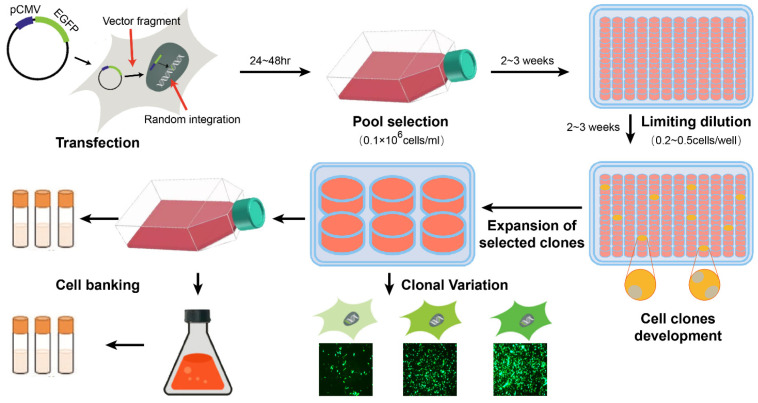
CHO cell line development (CLD) procedures.

**Table 1 ijms-26-01324-t001:** Summary of three transgene integration patterns.

Integration Pattern	Characteristics	Advantages	Disadvantages	References
Random transgene integration (RTI)	Random integration of transgenes onto the host cell chromosome	Operated simplify; can generate the highly expressed cell clones; gold standard	Laborious, inefficient, tedious; relies on chance, heterogeneity of cell clone; DNA double strand breaks; point mutations and sequence variants; concatemers	[9,10,11,12]
Semi-targeted integration (STI)	Transposase-mediated transgene semi-targeted integration, including Sleeping Beauty, PiggyBac, Leap in ^®^, and DirectedLuck ™ Transposition enzyme system	Does not occur within a predefined genomic context; high expression level and stability; shorter time frame	The integrated copy number is unpredictable; heterogeneity of cell clone	[6,13,14,15]
Site-specific integration (SSI)	Using enzyme integration, homologous directed repair and non-homologous end joining,	Site-specific integration; no heterogeneity of cell clone	Identification of a highly active and suitable locus; requires extensive experimental screening; low copy number, low expression level	[16,17,18,19,20]

**Table 2 ijms-26-01324-t002:** Intracellular causes and overcoming strategies for recombinant protein aggregation.

Causes	Mechanism	Overcoming Strategies	Results	References
Inefficiencies of ER	The ER ability to fold, process, and assemble proteins leads to the excessive misfolded proteins, forming aggregation; Delayed transport or insufficient amount of cargo receptors	Overexpression of ER-resident chaperones, such as PDI, BiP, Calnexin, Ero1l, XBP-1, ATF4, ATF6, ERGIC-53, MCFD2, Hsp105	Increased production and reduced aggregation	[16,73,74,75,76,77,78,79,80,81,82]
Molecular structures	The intrinsic properties of protein, gene mutations, transcription or translation errors, collapse of folding and chaperone mechanisms, errors in PTMs, protein transport, leading to protein misfolding	Sequence changes, such as Cys to Thr or Asn; Computer Design such as CrossMab; orthogonal Fab interface; tcBsIgG; LUZ-Y; OAscFab IgG technology	Increased production and reduced aggregation	[65,68,83,84,85,86,87,88,89,90,91,92,93]
Unequal expression or insufficient secretion	Inappropriate ratio of light chain (LC) to heavy chain (HC)	Changing the ratio of LC to HC, optimization of signal peptide	Increased production and reduced aggregation	[94,95,96]
ER autophagy dysfunction	ER autophagy eliminate unfolded protein reactions, facilitating the correct folding of proteins	Overexpression of anti-autophagy molecular, such as AKT1S1 and ATG16L1	Increased production and reduced aggregation	[16,97,98,99]

**Table 3 ijms-26-01324-t003:** Protein folding, secretion and aggregation regulator during cell culture.

Type	Name	Product/Condition	Effect	References
Culture medium components or additives	Tyrosine	CD20 antibody	High concentrations lead to the production of high aggregation	[103]
	Proline	COMP-Ang1	Increased production and reduced aggregation	[86,104]
	Trehalose	Bispecific antibody	Increased production and reduced aggregation	[105]
	Glycerol	FCA1 M-CSF	Increased production and reduced aggregation	[104]
	Dimethyl sulfoxide	COMP-Ang1 mAb	Increased production and reduced aggregation	[104]
	Sodium 4-phenylbutyrate	CA267T mutant	Improved protein secretion	[106]
	Sodium butyrate	mAb	Improved qp and productivity, reduced aggregation	[107]
	Choline	mAb	Increased production and decreased aggregation	[107]
Culture environment	Temperature (Standard: 37 °C)	IFN-β/30 °C, 32 °C	Increased production and decreased aggregation	[108]
		mAb/31.5, 32.5, 36 °C	Increased aggregation	[109]
		IgG-fusion/32 °C, 34 °C	Increased aggregation	[110]
		TNFR-Fc/31 °C	Decreased aggregation	[86]
	pH (Standard: 7.0)	FCA1/7.5	Decreased aggregation	[110]
	Osmolality (Standard: 280–340 mOsm/kg)	COMP-angiopoietin-1/400 mOsm/kg	Decreased aggregation	[111]
		IFN-β/470 mOsm/kg	Decreased aggregation	[112]
	pO2 (Standard: 50%)	IgG-fusion/15%	Decreased aggregation	[110]

## Data Availability

No new data were created or analyzed in this study.

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
