# Peer review of "From Cell Clones to Recombinant Protein Product Heterogeneity in Chinese Hamster Ovary Cell Systems"

_ijms, 2025, doi:10.3390/ijms26031324_

Round 1

Reviewer 1 Report

Comments and Suggestions for Authors

In their review Wang et al. discuss the problem of recombinant protein production.

They focused on Chinese hamster ovary cell system and discuss various pitfalls occurring during construction of the production clone and final protein production. Potential problems are identified, described in details and finally possible solutions are discussed.

In my opinion, authors discuss all main problems usually occurring during protein expression in cell-based systems and I thus have no major complaints.

Only minor comment is regarding the font used in the Tables. The font is rather blurred, indicating the tables were copied from another source. For the final version, I would check the print quality.

Comments on the Quality of English Language

Unfortunately, English language is the main problem of the review.

It was possible to basically understand the text, but many sentences need to be rephrased correctly. In many cases adverbs are not used appropriately, prepositions and articles are frequently misused.

Author Response

Comments and Suggestions for Authors

In their review Wang et al. discuss the problem of recombinant protein production.

They focused on Chinese hamster ovary cell system and discuss various pitfalls occurring during construction of the production clone and final protein production. Potential problems are identified, described in details and finally possible solutions are discussed.

In my opinion, authors discuss all main problems usually occurring during protein expression in cell-based systems and I thus have no major complaints.

Only minor comment is regarding the font used in the Tables. The font is rather blurred, indicating the tables were copied from another source. For the final version, I would check the print quality.

Answer: Thanks for your comment.We have revised the Tables.

Comments on the Quality of English Language

Unfortunately, English language is the main problem of the review.

It was possible to basically understand the text, but many sentences need to be rephrased correctly. In many cases adverbs are not used appropriately, prepositions and articles are frequently misused.

Answer: Thanks for your comment. We have rechecked typo and grammatical errors and asked editage Company to help us to edit the language once again. The following file is the “CERTIFICATE OF ENGLISH EDITING” from them.

Reviewer 2 Report

Comments and Suggestions for Authors

The manuscript “From Cell Clone to Recombinant Protein Product Heterogeneity in the Chinese Hamster Ovary Cell System” is a review that explores the technological approaches used to produce recombinant therapeutic proteins using Chinese Hamster Ovary (CHO) cells. The review presents three main systems for generating recombinant cell clones: Random Transgene Integration, Semi-targeted Integration, and Site-specific Integration. Based on these approaches, the authors discuss the differences in transgene integration patterns within the CHO cell genome, clone heterogeneity, and the challenges associated with transgene expression.

The manuscript is well-written, logically structured, and provides a comprehensive overview of information valuable for both academic and industrial applications in mammalian protein production. To further enhance the collected data, it would be beneficial to include a brief paragraph discussing scale-up technologies and how these procedures can impact or optimize transgene expression. Additionally, the authors should address typographical and grammatical errors, such as the incorrect use of punctuation marks (e.g., in lines 210-224).

Comments on the Quality of English Language

While the English is understandable, thorough review is suggested.

Author Response

The manuscript “From Cell Clone to Recombinant Protein Product Heterogeneity in the Chinese Hamster Ovary Cell System” is a review that explores the technological approaches used to produce recombinant therapeutic proteins using Chinese Hamster Ovary (CHO) cells. The review presents three main systems for generating recombinant cell clones: Random Transgene Integration, Semi-targeted Integration, and Site-specific Integration. Based on these approaches, the authors discuss the differences in transgene integration patterns within the CHO cell genome, clone heterogeneity, and the challenges associated with transgene expression.

The manuscript is well-written, logically structured, and provides a comprehensive overview of information valuable for both academic and industrial applications in mammalian protein production. To further enhance the collected data, it would be beneficial to include a brief paragraph discussing scale-up technologies and how these procedures can impact or optimize transgene expression. Additionally, the authors should address typographical and grammatical errors, such as the incorrect use of punctuation marks (e.g., in lines 210-224).

Comments on the Quality of English Language

While the English is understandable, thorough review is suggested.

Answer: Thanks for your comment. We have revised the incorrect use of punctuation marks in lines 210-224. We also have rechecked typo and grammatical errors and asked editage Company to help us to edit the language once again. The following file is the “CERTIFICATE OF ENGLISH EDITING” from them.